

# Phytogeographic and genetic variation in *Sorbus*, a traditional antidiabetic medicine—adaptation in action in both a plant and a discipline

Anna Bailie[1,2], Sebastien Renaut[3], Eliane Ubalijoro[4], José A. Guerrero-Analco[5,6], Ammar Saleem[5], Pierre Haddad[7], John T. Arnason[5], Timothy Johns[1,8] and Alain Cuerrier[2]

[1] Department of Plant Science, McGill University, Ste-Anne-de-Bellevue, QC, Canada
[2] Jardin Botanique de Montréal, Institut de Recherche en Biologie Végétale, Montreal, QC, Canada
[3] Département de Sciences Biologiques, Institut de Recherche en Biologie Végétale, Université de Montréal, Montreal, QC, Canada
[4] Institute for the Study of International Development, McGill University, Montreal, QC, Canada
[5] Department of Biology, University of Ottawa, Ottawa, ON, Canada
[6] Red de Estudios Moleculares Avanzados, Instituto de Ecología A.C, Xalapa, Veracruz, Mexico
[7] Département de Pharmacologie, University of Montreal, Montreal, QC, Canada
[8] School of Dietetics and Human Nutrition, McGill University, Ste-Anne-de-Bellevue, QC, Canada

Corresponding authors
Sebastien Renaut,
sebastien.renaut@gmail.com
Alain Cuerrier,
alain.cuerrier@umontreal.ca

## ABSTRACT

Mountain ash (*Sorbus decora* and *S. americana*) is used by the Cree Nation of the James Bay region of Quebec (Eeyou Istchee) as traditional medicine. Its potential as an antidiabetic medicine is thought to vary across its geographical range, yet little is known about the factors that affect its antioxidant capacity. Here, we examined metabolite gene expression in relation to antioxidant activity, linking phytochemistry and medicinal potential. Samples of leaf and bark from *S. decora* and *S. americana* were collected from 20 populations at four different latitudes. Two genes known to produce antidiabetic substances, flavonol synthase and squalene synthase, were analyzed using quantitative real time PCR. Gene expression was significantly higher for flavonol synthase compared to squalene synthase and increased in the most Northern latitude. Corresponding differences observed in the antioxidant capacity of ethanolic extracts from the bark of *Sorbus* spp. confirm that plants at higher latitudes increase production of stress-induced secondary metabolites and support Aboriginal perceptions of their higher medicinal potential. Modern genetic techniques such as quantitative real time PCR offer unprecedented resolution to substantiate and scrutinise Aboriginal medicinal plant perception. Furthermore, it offers valuable insights into how environmental stress can trigger an adaptive response resulting in the accumulation of secondary metabolites with human medicinal properties.

## INTRODUCTION

Indigenous people have used plants as medicine for thousands of years (*Arnason, Hebda & Johns, 1981*). They also have developed a number of non-technological approaches to identify plants and locations that will have the greatest concentration of medicinal active compounds (*De Almeida, De Amorim & De Albuquerque, 2011*). Modern experimental methods have confirmed these traditional approaches (*Haddad et al., 2012*; *Rapinski et al., 2014*; *Rapinski et al., 2015*) and have helped to identify useful phytochemicals compounds (*Shang et al., 2012*). Nevertheless, there remains a great divide between medicinal plant information held by indigenous groups and what is known from modern scientific inquiry.

The Canadian Institutes of Health Research (CIHR) Team on Aboriginal Antidiabetic Medicines (TAAM) studies medicinal plants used in Cree communities for symptoms of Type 2 Diabetes (T2D) (*Leduc et al., 2006*; *Fraser et al., 2007*; *Cuerrier et al., 2012*). Type 2 Diabetes is reaching epidemic levels among Aboriginal groups in different regions of the world (*Australian Institute of Health Welfare, 2015*), including the Cree of Eeyou Istchee (James Bay, Quebec: see *Leduc et al., 2006*; *Young et al., 2000*) and there is an urgent need to address this issue. Ethnobotany has been used to document, describe and explain complex relationships between human societies and their use of plants. In addition, ethnobotany, has taken advantage of new technologies and methodologies (*Leduc et al., 2006*; *Cuerrier et al., 2015*; *Carrasco et al., 2016*) to confirm and translate traditional concepts into scientific language.

Here, ethnobotanical interviews with Cree healers and elders in addition to laboratory analyses have helped to identify *Mushkuminanatikw (Sorbus* spp.) among a number of plant species possessing putative antidiabetic properties (*Leduc et al., 2006*; *Spoor et al., 2006*; *Nachar et al., 2013*; *Shang et al., 2015*). Specifically, two species of *Sorbus* (Rosaceae, *S. decora* (Sargent) C.K. Schneider [Showy mountain ash] and *S. americana* Marshall [American mountain ash] have been shown to have glucose mediating activity *in vivo* (*Vianna et al., 2011*), with other diabetes-related activity demonstrated *in vitro* (*Spoor et al., 2006*; *Nachar et al., 2013*; *Shang et al., 2015*). As such, they represent prime candidates to help treat T2D.

These two species also exhibit differences in antioxidant capacity according to locality and tissue type (*Fraser et al., 2007*). Among these differences, previous samples collected from Whapmagoostui, a coastal community in the north of the Eeyou Istchee region showed a higher capacity to reduce oxidative molecules in a number of assays, as compared to those from Mistissini, a southern inland community (*Fraser et al., 2007*). Accordingly, there is a general perception among Cree healers and elders that plants from Northern latitude and coastal regions are more efficient as traditional medicine. Furthermore, differences in phenolic compounds and activities observed according to tissue type were supported by traditional Crees, which use decoction of specific parts of the plant for different symptoms (*Fraser et al., 2007*). Inner bark, which is preferred medicinally, tends to show higher antioxidant capacity than leaves (*McCune & Johns, 2002*; *Fraser et al., 2007*).

Secondary metabolites, including those with antioxidant activity, are known to aid plants in dealing with environmental stresses, such as photoperiod, temperature, pedology and other geomorphological parameters (*McCune & Johns, 2007*; *Theis & Lerdau, 2003*;

*Dixon & Paiva, 1995*; *Figueiredo et al., 2008*). Yet relevant classes of metabolites, and their expression, have not been properly studied with respect to ethnobotanical questions (*Castellarin et al., 2006*; *Hayashi et al., 2004*). Elucidating the links between metabolite gene expression and phytochemistry-related function can help explain inter- and intra-specific variation of antioxidant properties across both geographical and climatological gradients. Antioxidant compounds generally accumulate in response to various stresses encountered by the plants, including such oxidizing pressures as UV-B radiation (*Figueiredo et al., 2008*). For example, *Stushnoff & Junttila (1986)* hypothesized that trees growing in Northern areas will experience greater environmental stress due to the greater range of temperature they must face. Similarly, coastal plants that cope with greater amounts of salt in their soils could also face greater environmental stress. Consequently, both Northern and coastal plants would cope with these stressors by producing more secondary metabolites, which may in turn have medicinal value.

For a plant of ethnomedicinal importance, phenotypic variation in the quantity and quality of useful phytochemicals can determine its biological capacity to mitigate disease. Recent studies have shown that plant phenolic metabolites have high antioxidant activity and certain therapeutic properties, including antidiabetic and anti-hypertension activity (*Kwon, Vattem & Shetty, 2006*). Analytical techniques such as ORAC (oxygen radical absorbance capacity) and DPPH (2,2-Diphenyl-1-Picrylhydrazyl) assays can facilitate the determination of the biological capacity of constituents attributed to the health benefits of functional foods such as berries and potential traditional medicines (*Hukkanen et al., 2006*; *McCune & Johns, 2007*; *Harris et al., 2014*). In addition, the expression of genes involved in the production of secondary metabolites with antioxidant activity might be accurate indicators of the medicinal quality of the plants, and thus should vary depending on eco/geographical parameters. In some instances, genetic differences may underlie the production of secondary metabolites and thus be locally adaptive, while concomitantly, phenotypic plasticity could also lead to differences in the production of secondary metabolites. At the present time, these questions have been subject to only preliminary inquiry for plants of medicinal and ethnobotanical interest (*Bottin et al., 2007*).

Testing for variation of *Sorbus* spp. across populations can help to determine if differences in expression are due to genetics or adaptation to surrounding environments. Finally, taking advantage of modern genetic techniques such as rt-PCR allows ethnobotany to evolve and use new tools to document specific traditional knowledge (*Cordell, 2011*). With ongoing losses of biodiversity, there is an urgent need to explore plants using all possible tools to learn as much as possible about them, before important medicinal compounds are lost (*Newmaster, Ragupathy & Janovec, 2009*).

In this paper, we measured levels of gene expression for two enzymes involved in the biosynthesis of important secondary metabolites known to have antidiabetic properties in two species of *Sorbus* (flavonol synthase, squalene synthase). We measured gene expression using quantitative real time PCR in order to see how it varied according to tissue types and sample localities (latitude and longitude) and hypothesized that they would be more highly expressed in Northern and coastal populations. In this, we follow the Cree elders who discussed with our team the importance of these populations for medicinal purposes.

Therefore, our main goal is to document, and to a certain extent, support Cree knowledge about where best medicinal plants can be found within their territory. In addition, we measured antioxidant activity according to tissue types and sample localities (latitude and longitude) using both oxygen radical absorbance capacity (ORAC) and 2,2-Diphenyl-1-Picrylhydrazyl (DPPH) assays. We hope to continue bridging this gap using *Sorbus* species as a model system given its well-known antidiabetic potential.

## METHODS

### Field collection

Samples were collected in August 2008 in northern Quebec (Canada) within the James Bay territory, which is independently administered by the Cree Nation Government and the local Band councils of each community. Twenty populations were sampled in the areas surrounding the villages of Waskaganish (51°29′N, 78°45′W), Chisasibi (53°47′N, 78°53′W) and Whapmagoostui (55°16′N, 77°45′W) on the coast of James Bay and Hudson Bay, and Nemaska (51°41′N, 75°15′W), Mistissini (50°25′N, 73°51′W) located inland. All the villages, except Whapmagoostui, are in the boreal forest ecozone, with spruce and other coniferous trees dominating. Whapmagoostui is located in the hemiarctic, where the tundra ecozone begins. The field permit was covered within a Research Agreement among universities, the Cree Board of Health and Social Services of James Bay, and the participating communities (see http://www.taam-emaad.umontreal.ca/about%20us/CIHR-TAAM_Final_Research_Agreement_signed_091030.pdf).

For each of twenty populations, bark and leaves samples were collected for phytochemical extraction from two, and up to twenty, trees respectively. These were dried either in a plant press or in paper bags (approximately 10 g of dry material). Effort was taken to collect both *S. americana* and *S. decora* individuals in equal number, but *S. americana* was found to be less common or absent from Whapmagoostui. At each site, GPS location was recorded and several pictures of the surroundings were taken. All samples were collected between 10 AM and 2 PM. At each site, voucher specimens of both *S. decora* and *S. americana* were collected in duplicate for depositing at the Marie-Victorin Herbarium (MT) at the Montreal Botanical Garden and at the McGill Herbarium (MTMG) on the Macdonald Campus; identities were confirmed by S Bailleul, S Hay and A Cuerrier at the Montreal Botanical Garden.

Samples were also collected for RNA extraction from both leaf and bark tissues. Approximately 200 mg of bark and leaf samples were immediately immersed in 1 ml of RNAlater$^®$ (Qiagen, Venlo, Netherlands). This approach had been previously tested using samples collected from the Montreal Botanical Garden, and found to successfully maintain RNA integrity. Once collected, samples were placed in a −20 °C freezer for transfer between villages and back to the laboratory.

### RNA extraction, selection of candidate genes and primers

Two enzymes, flavonol synthase (FLS) and squalene synthase (SquaS), were selected given that both are involved in the biosynthesis of precursors of important secondary metabolites in *Sorbus* (*Fischer et al., 2007*) and possess well known antioxidants properties. Flavonol synthase is a key enzyme in the antioxidant flavonoid pathway (*Hukkanen et al., 2006*;
*Fischer et al., 2007*) and the gene expression of FLS has been demonstrated to control enzyme activity in many plants species (*Xu et al., 2012*). For example, in *Vitis vinifera*, gene expression of FLS increased during ripening of the fruit, coincident with the increase in flavonols per berry (*Downey, Harvey & Robinson, 2003*). Squalene synthase (SquaS) is an enzyme expressed early in the biosynthetic chain for terpenes, a large group of volatile unsaturated hydrocarbons found in the essential oils of plants and well known for their antioxidant capacities (*Gonzalez-Burgos & Gomez-Serranillos, 2012*). In addition, the positive correlation between SquaS gene expression, protein levels and enzymatic activity has been clearly demonstrated in *Nicotiana tabacum* (*Devarenne, Ghosh & Chappell, 2002*).

As no published sequences for FLS and SquaS from *Sorbus* were available, primers were designed based on an Expressed Sequence Tag (EST) sequence from *Malus* (GenBank Accession #AB331947 and #DQ849001, see Table S1). Actin gene expression (GenBank Accession served as a control for real time quantitative PCR, as commonly performed (*Heid et al., 1996*). Primers were designed based on *Sorbus aucuparia* Linnaeus sequences. Primers for all genes were designed in Primer3 (*Rozen & Skaletsky, 2000*) and synthesized by Alpha DNA (Montreal, Quebec). Primers were first tested by amplifying genomic DNA samples, which were then used to create purified PCR standards for all three genes (PCR Purification Kit, Qiagen). PCR products for the two candidate genes were sequenced by Genome Quebec and consensus sequences compared (BLASTn) against Expressed Sequences Tags (ESTs) database in GenBank (http://blast.ncbi.nlm.nih.gov/Blast.cgi, performed in July 2008). Both genes had hits against *Malus* ESTs (FLS *S. decora*, GenBank accession number GQ423752, FLS *S. americana* GenBank accession number GQ423751, SquaS *S. decora* GenBank accession number GQ423753, SquaS *S. americana* GenBank accession number GQ423754).

RNA was extracted following the protocol outlined in the Qiagen RNEasy Mini Plant Kit manual. The samples were extracted following the standard RNEasy protocol, with a standard (200 µl) amount of ethanol added to each sample, instead of approximately half the volume as recommended. At the end of the RNA extraction protocol, concentration and quality were verified using a NanoDrop spectrophotometer (Thermo Scientific, Waltham, Ma, USA). Ten samples out of the 78 collected had a 260/280 ratio below 1.6, or a low yield of total RNA, were not processed any further. Using the Wipeout Qiagen Quantitect reverse transcriptase protocol, cDNA was synthesized, with a standard amount of 40 ng of RNA as starting material.

## Quantitative RT-PCR

RT-PCR was performed on a Stratagene (Agilent Technologies, Santa Clara, CA, USA) MX3000P real-time PCR system with a Brilliant SYBR Green rtPCR Master Mix kit. A PCR performed against actin provided a base value, which was then compared to the purified genomic PCR products of each gene, and then diluted to a concentration between $10^{-9}$ and $10^{-12}$ of the original product (QiaQuick, Qiagen). Products were run in strip tubes of eight, with standards in every reaction, and then samples were randomized in rows of gene of interest, followed by the control gene (actin). It took three sets of samples to complete one repeat of each gene, and all sets were run three times. Amplification conditions were as follows for FLS: denaturing step (10 min at 95 °C), followed by 45 cycles

of 95 °C denaturation (30 s), 55 °C annealing (1 min), 72 °C extension (30 s), completed by a final cycle of 1 min at 95 °C, 30 s at 55 °C and 30 s at 95 °C. For squalene synthase, amplification conditions were as follows: denaturing step (10 min at 95 °C), followed by 50 cycles of 95 °C denaturation (30 s), 53 °C annealing (1 min), 72 °C extension (30 s), completed by a final cycle of 1 min at 95 °C, 30 s at 53 °C and 30 s at 95 °C.

The slopes of product accumulation for the genes of interest against the standards were checked to be between −3.5 and −4.5, along with the efficiency of the reaction being between 90 and 110%, and a singular dissociation curve. Final copy value numbers, based on the standard curve of each reaction, the $C(t)$ of each sample and the efficiency of the reaction were exported to a spreadsheet. Copy values for each sample were normalized for each gene of interest against actin. Each sample was analysed three times, and outliers were identified using the Grubbs test (*Grubbs, 1950*) and then removed using Prism GraphPad (San Diego, CA, USA). Gene expression data are available as Table S2.

We analysed gene expression differences separately for flavonol and squalene synthase using linear mixed effect models in R (*R Core Team, 2013*). First, expression ratios (FLS/actin or SquaS/actin expression) were log transformed. Several linear mixed effect models (package lme4, *Bates et al., 2015*) with a combination of fixed effects (species, latitude, longitude, tissue) and random effects (individuals, replicates) were tested. The model with the lowest Akaike information criterion (AIC, *Akaike, 1974*) was retained (see Table S3 for list of models tested; *Burnham & Anderson, 2004*). *F* statistics and *p*-values based on Satterthwaite's approximations were calculated with LmerTest (*Kuznetsova, Christensen & Brockhoff, 2013*) in R. Post hoc Tukey *t*-tests were also performed in R.

## Extract preparation

A total of 24 bark samples based on sample availability, from all five communities and four latitudes, were extracted and analysed for their antioxidant activity and phenolic content. For extraction, approximately 5 g of dried material was ground using a blender (40 mesh), placed in an Erlenmeyer flask with 100 ml of 80% ethanol, sealed and swirled for 24 h at approximately 150 rpm. The filtrate was separated from residue and the pellet was extracted twice. Pooled filtrate (200 ml) was dried using a rotary evaporator (water bath temperature of 40 °C), freeze dried in Supermoduylo freeze drier (Thermo Fisher Inc.) and stored at –20 °C.

## Antioxidant analysis

Stock concentrations made from the crude bark extracts were analyzed for antioxidant activity using both oxygen radical absorbance capacity (ORAC) and 2,2-Diphenyl-1-Picrylhydrazyl (DPPH) assays.

ORAC assays were based on a protocol designed by *Cao & Prior (1999)* and adapted by *Ou, Hampsch-Woodill & Prior (2001)*. It uses fluorescein reacting with AAPH, a peroxyl radical, to measure an extract's ability to protect against oxidation compared to Trolox, a known antioxidant analogous to vitamin E and serving as positive control. Several modifications were made to the *Ou, Hampsch-Woodill & Prior (2001)* protocol. Each extract (0.005 g) was dissolved in 25 ml of 50% methanol and potassium phosphate buffer

(PBS, pH 7.0), while Trolox, AAPH and fluorescein, prepared fresh for each reaction, were diluted in 100% PBS. Trolox was diluted to four concentrations, 0.78 μM, 0.39 μM, 0.195 μM and 0.0975 μM. Extracts were diluted to 6.5 μg/ml, 3.25 μg/ml, 1.6125 μg/ml and 0.806125 μg/ ml. Each reaction took place at room temperature and included 150 μl of fluorescein and 25 μl of each concentration of Trolox or sample; 25 μl of AAPH was added to each well at the start of the reaction. A fluorescence reading was taken as soon as AAPH was added to the 96-wells plate; readings were then taken every 10 min for 90 min Each concentration of sample or Trolox was conducted in duplicate for every reaction, with the reaction being repeated three times. Areas under the curve for each concentration were calculated in a spreadsheet with the blank subtracted from each sample area under the curve (AUC). Final values were determined using a regression equation based on Trolox curves, expressed as micromoles Trolox equivalents per gram.

DPPH assays were conducted based on a protocol designed by *Blois (1958)* with modifications outlined in *McCune & Johns (2002)*, *Owen & Johns (2002)* and *Fraser et al. (2007)*. The free radical scavenging activities of extracts when exposed to DPPH (1,1-diphenyl-2-picrylhydrazyl; Sigma-Aldrich, Oshawa, Ontario, CAN) for 10 min were measured using a Wallac Victor[2] 1420 Multilabel counter (PerkinElmer, Waltham, Ma, USA), with an absorbance of 490 nm. An $IC_{50}$ (inhibition concentration based on concentration of ascorbic acid required to quench 50% of radicals in the reaction) was calculated for each sample based on the linear portion of the dose–response curve for samples. The amount quenched by the standards for comparison to samples was calculated using both the linear and plateau portions of the line, using the equation:

$$\hat{y} - y_0 = \frac{\sum \{(x - x_0)(y - y_0)\}}{\sum \{x - y_0\}^2}(x - x_0).$$

A smaller value thus denotes a higher free radical scavenging ability. Reactions were performed three times, with every sample in duplicate within each reaction. Catechin, epicatechin and quercetin served as controls along with ascorbic acid. Statistical analysis (ANOVAs and post hoc Tukey $t$-tests) for both ORAC and DPPH data sets was performed in SPSS (v 16.0).

Finally, total soluble phenolic content of each bark sample ($n = 72$) was determined using the Folin-Ciocalteu method as described by *Fraser et al. (2007)* and significance tested with non-parametric Kruskal–Wallis test and post hoc Tukey $t$-tests.

## RESULTS

### RNA analysis

Of the 78 collected samples, 59 were successfully extracted and transcribed into high quality cDNA, suitable for gene expression analyses. It is likely that the high phenolic content of *Sorbus* made it difficult to use large amounts of RNA suitable for reverse transcription (*Kiefer, Heller & Ernst, 2000*). We observed that flavonol synthase gene expression was consistently higher than squalene synthase ($t$-test, $t = 11.1$, $p$-value $= 3.14e-24$, Fig. 1).

For both flavonol and squalene synthase, a model containing latitude as a fixed factor was chosen (lowest AIC value) and expression differed significantly according to latitude

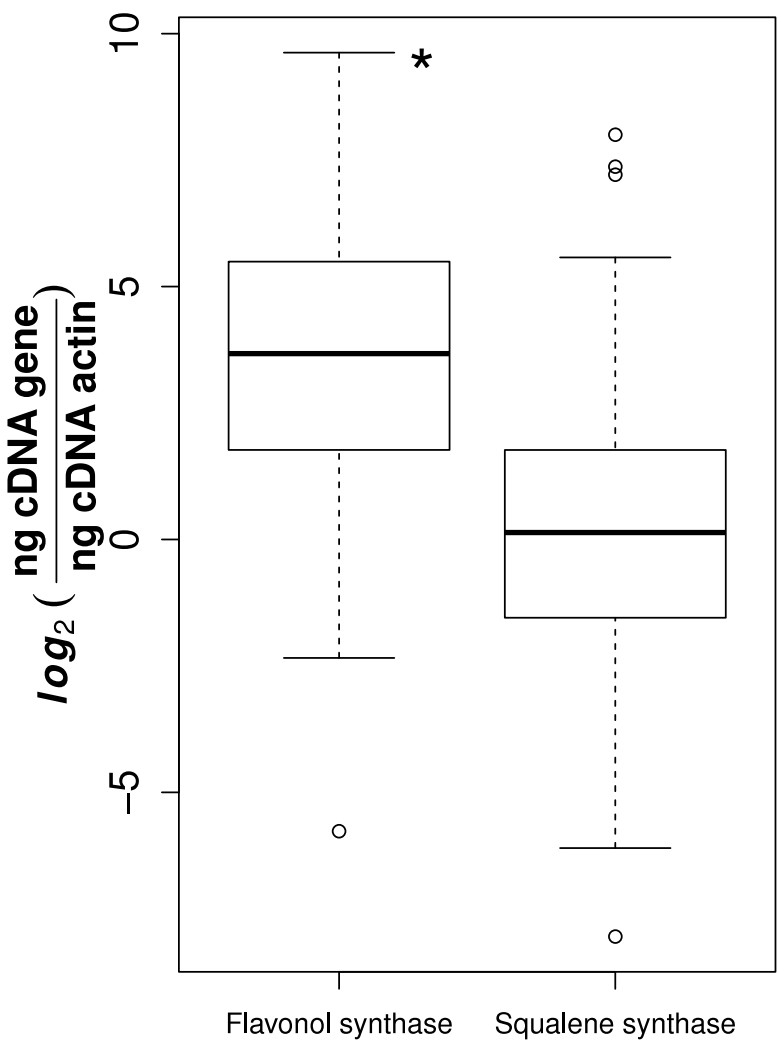

**Figure 1  Gene expression for flavonol and squalene synthase.** Asterisk represents significant difference between groups (*t*-test, *p*-value = 3.14e−24).

(*F* statistic = 3.48, *p*-value = 0.029 and *F* statistic = 4.02, *p*-value = 0.015, respectively). In both cases, an increase in expression according to latitude was observed (Figs. 2A and 2B) and post hoc Tukey tests indicated that at latitude 55, expression was significantly higher. For both squalene and flavonol synthase, we also observed that gene expression in coastal samples was higher than those from inland locales in the full mixed effects model that contained all variables including longitude, but this effect was non-significant (Table S3, squalene synthase, *F* statistic = 0.34, *p*-value = 0.56; flavonol synthase, *F* statistic = 0.36, *p*-value = 0.55). Similarly, we observed non-significant differences in the full model between gene expression in leaf and bark samples for both squalene synthase (Table S3, *F* statistic = 0.28, *p*-value = 0.60) and flavonol synthase (Table S3, *F* statistic = 1.07, *p*-value = 0.30).

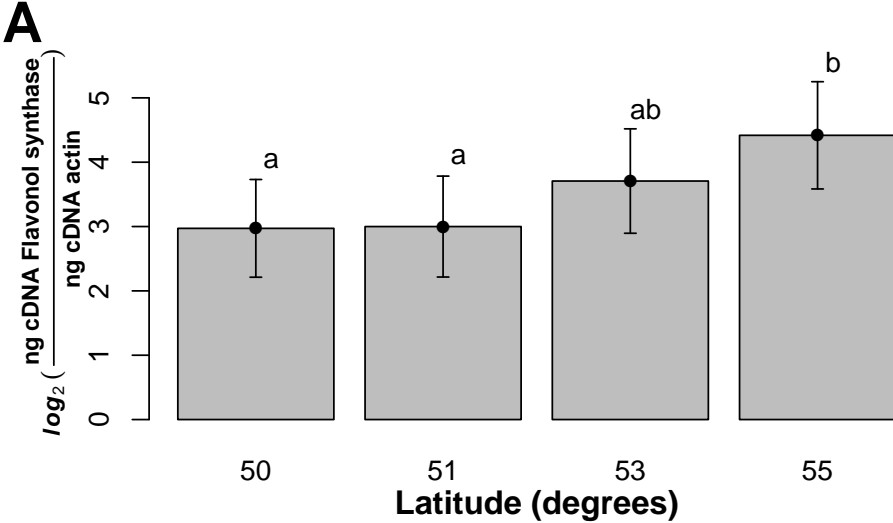

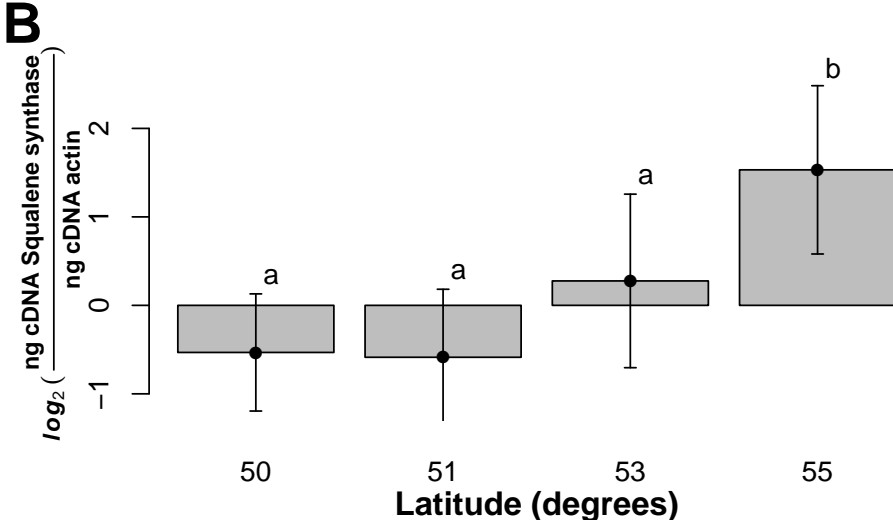

**Figure 2** Gene expression with respect to latitude (bars represent 95% CI) for (A) Flavonol synthase and (B) Squalene synthase.

## Antioxidant analysis

Both ORAC and DPPH analyses showed similar trends in antioxidant capacities of samples pooled according to location ($r^2 = 0.92$, $p$-value $= 0.04$), although ANOVAs did not reveal significant differences ($p$-value $> 0.05$; Figs. 3A and 3B). Both analyses showed increasing antioxidant capacity with latitude (note that for DPPH, lower values imply greater antioxidant activity, Fig. 3A) similar to the trend observed with levels of gene expression for FLS ($r^2_{(ORAC-FLS\ expression)} = 0.90$, $p$-value $= 0.05$) and SquaS ($r^2_{(ORAC-SquaS\ expression)} = 0.84$, $p$-value $= 0.08$). *Sorbus decora* samples tended to show higher capacity than *S. americana* (Figs. 4A and 4B). It was also observed that samples collected from coastal locations showed a higher capacity compared to inland populations,

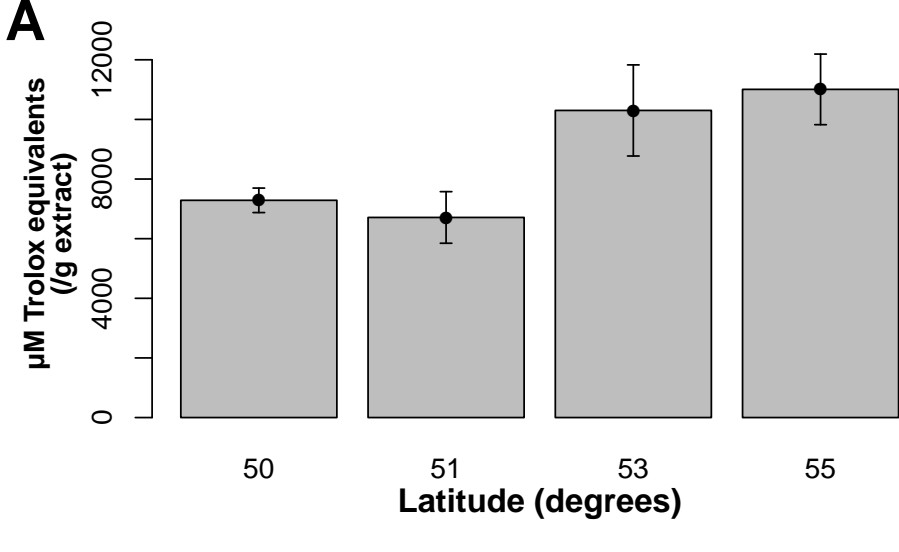

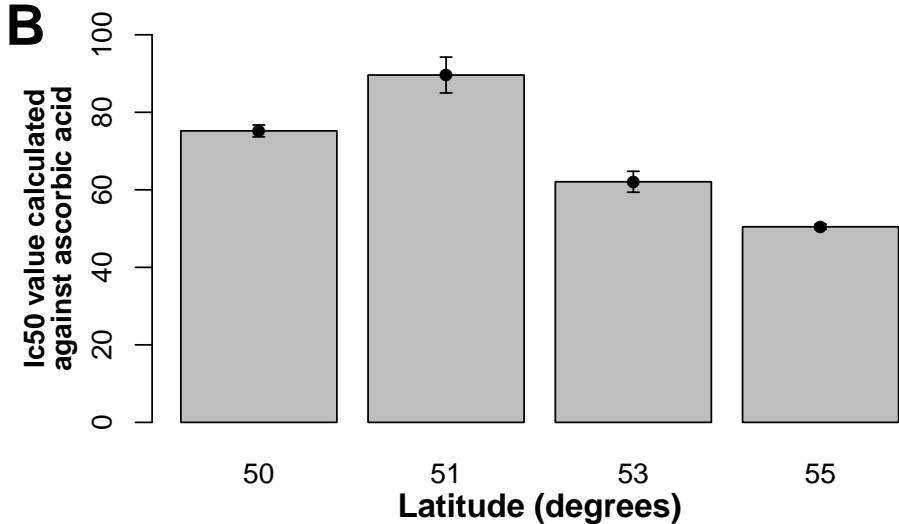

**Figure 3** (A) Antioxidant capacities shown as Trolox equivalents for phytochemical samples extracted from bark of *S. decora* and *S. americana* (pooled) collected from four latitudes, determined through ORAC. Bars represent the mean of duplicate samples assayed in triplicate ($\pm$SE). (B) Antioxidant capacities depicted as IC$_{50}$ values, for phytochemical samples extracted from bark of *S. decora* and *S. americana* (pooled) collected from four latitudes, determined through DPPH. Bars represent the mean of duplicate samples assayed in triplicate ($\pm$SE).

non-significantly in the ORAC assay but significantly in DPPH analysis (two-tailed $t$-test, $p$-value $= 0.03$, Figs. 5A and 5B).

Using the Folin-Ciocalteu method of phenolics measurement, we observed that soluble chemical content increased significantly with latitude (Kruskal–Wallis test, $p$-value $\leq$ 0.05, Fig. 6), along with significantly greater phenolic content in coastal samples (costal = 142.2, inland = 109.2, $p$-value $= 0.02$) and *S. decora* compared to *S. americana*, but not significantly (132.4 vs. 124.5 respectively, two-tailed $t$-test, $p$-value $= 0.6$).

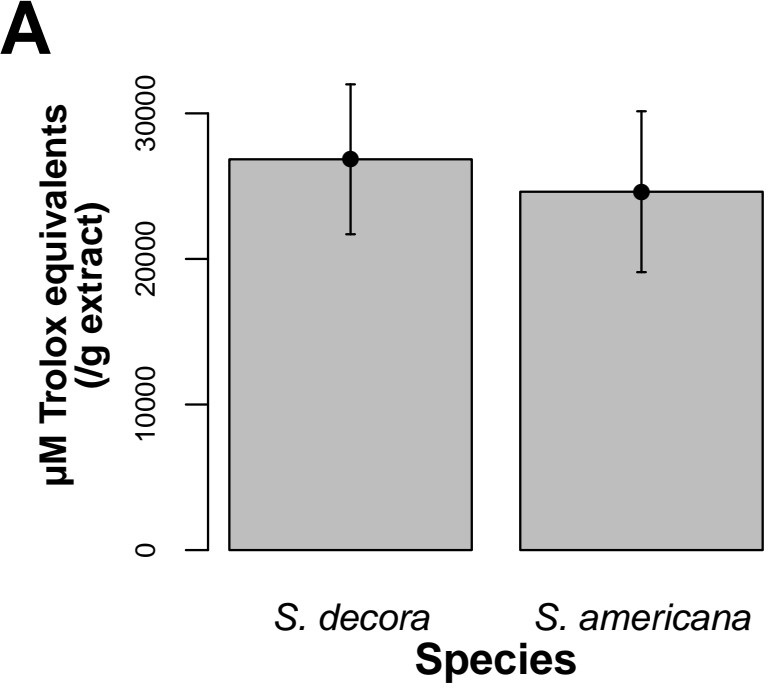

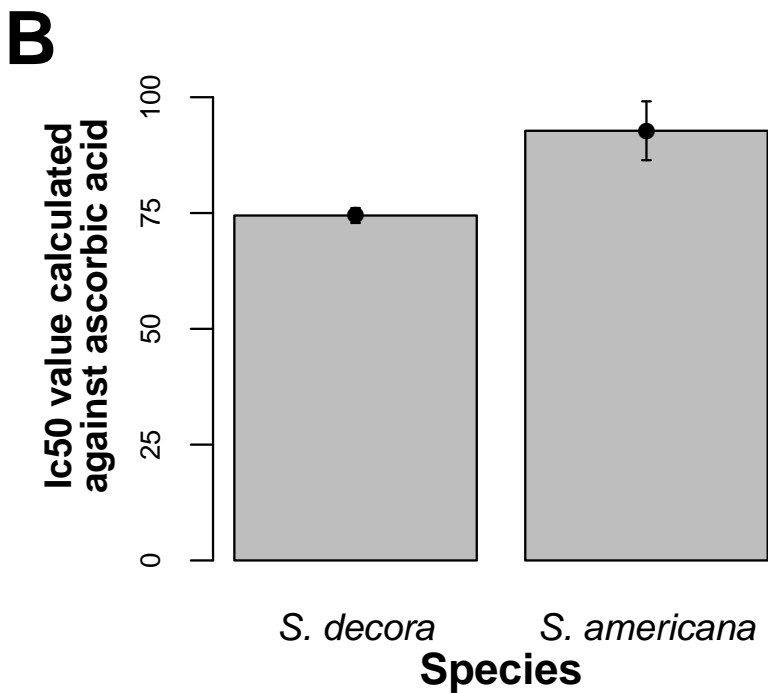

**Figure 4** (A) Antioxidant capacities shown as Trolox equivalents for both *S. decora* and *S. americana* bark samples, determined through ORAC. Bars represent the mean of duplicate samples assayed in triplicate (±SE). (B) Antioxidant capacities depicted as $IC_{50}$ values, for both *S. decora* and *S. americana* samples, determined through DPPH. Bars represent the mean of duplicate samples assayed in triplicate (±SE).

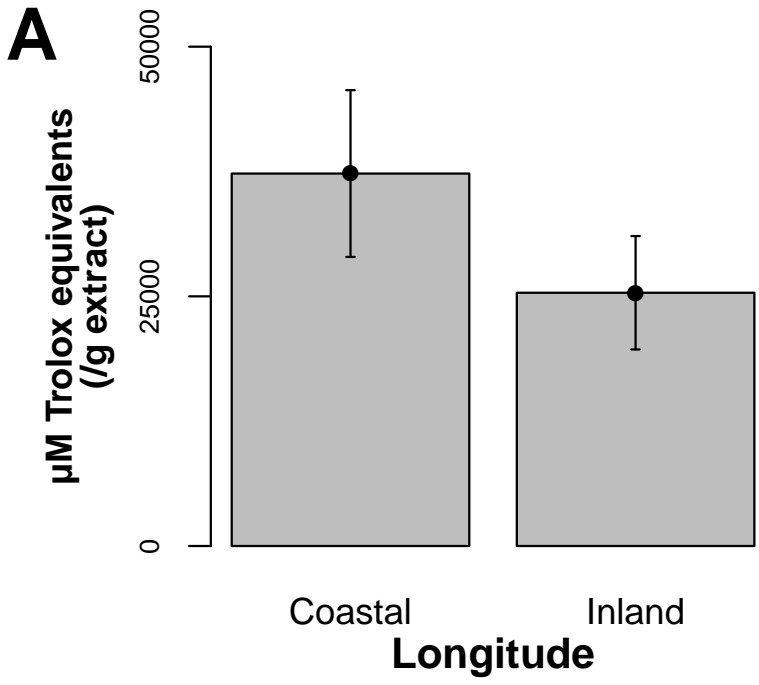

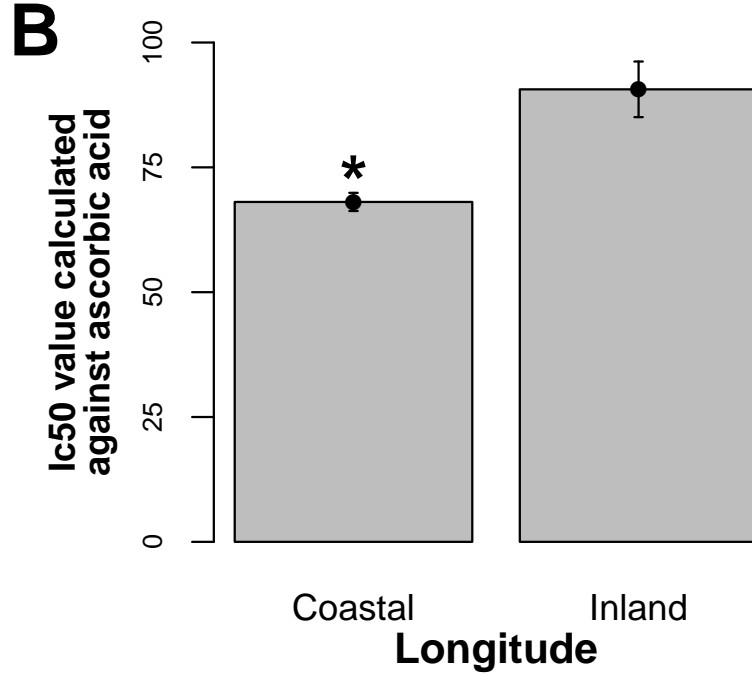

**Figure 5** (A) Antioxidant capacities shown as Trolox equivalents for coastal and inland phytochemical bark samples, (both *S. decora* and *S. americana*) determined through ORAC. Bars represent the mean of duplicate samples assayed in triplicate (±SE). (B) Antioxidant capacities depicted as $IC_{50}$ values, for coastal and inland phytochemical samples (both *S. decora* and *S. americana*), determined through DPPH. Bars represent the mean of duplicate samples assayed in triplicate (±SE). Asterisk represents significant difference between groups (two-tailed *t*-test, *p-value* = 0.03).

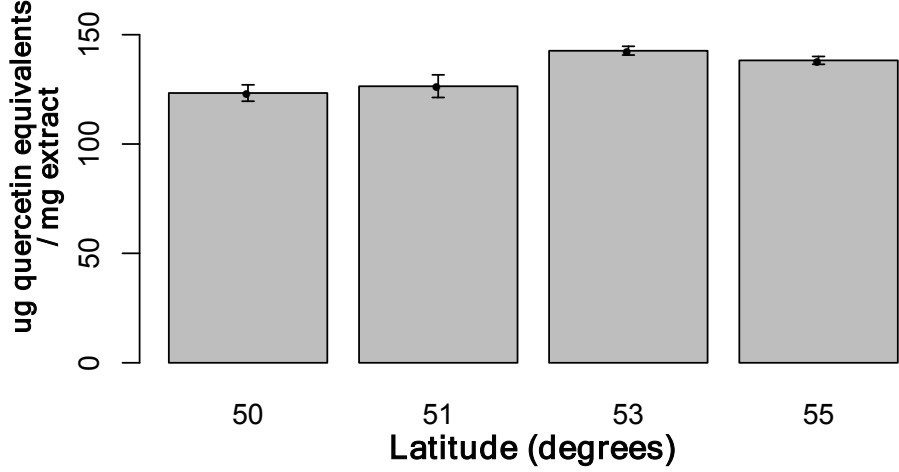

**Figure 6** **Total soluble phenolic content of bark samples ($n$ = 72) determined using the Folin-Ciocalteu method as described by** *Fraser et al. (2007).* Bars represent the mean of each sample assayed in triplicate (±SE).

## DISCUSSION

One of the central dogma in molecular biology states that DNA is transcribed into mRNA, and the latter translated into proteins, which then carry a vast number of metabolic function. As such, measures of gene expression revealing functional differences underlie a vast body of literature (*Vandesompele et al., 2002*; *Wang, Gerstein & Snyder, 2009*), and several studies have shown the correlation between mRNA-protein-enzyme activity (e.g., *Maier, Güell & Serrano, 2009*). Clearly, this assumption oversimplifies the true complexity of biological systems, owing to several reasons (e.g., post-transcriptional gene silencing, post-translational protein modification, experimental noise). Consequently, we firmly believe that it is only through the integration of different lines of evidence (traditional knowledge from the Cree healers, knowledge gained from gene expression, antioxidants and phenolic content experiments) that a clearer picture can emerge and solution to the T2D epidemic can be reached.

When our team discussed harvesting medicinal plants within the Eeyouch territory, Cree Elders and active members of communities involved in our project actively reported during meetings that plants are not always alike medicinally or otherwise. Harvesting should occur where plants show the highest medicinal activity. Then, they added that plants from northern latitudes and near coastal regions ranked amongst the best locales for gathering. Following their advices, we created a number of small parallel projects of which this one is part of. Again, traditional ecological knowledge has proved to be correct. Equally, both pollution and climate change are important concerns to healers and elders of the Cree nations, suggesting that they interfere with plant abilities. Elevated soil salinity levels from the nearby Bays may also have an impact on the production of secondary metabolites and hence the medicinal activity.

Indeed, the general trend of greater gene expression in samples collected from higher latitude and to some extent coastal locations is consistent with divergent metabolic

responses stimulated by the environment (*Hartmann et al., 2005*; *Rapinski et al., 2014*; *Rapinski et al., 2015*). Both the increase at higher latitudes of the expression of genes involved in the production of secondary metabolites in bark and leaf tissue, and the increase in antioxidant activities and phenolic content for bark tissue at higher latitudes are supported by previous findings (*Fraser et al., 2007*; *McCune & Johns, 2007*; *Rapinski et al., 2014*; *Rapinski et al., 2015*). Plants growing at higher latitudes are theorized to cope with a greater stress such as UV radiation, a shorter growing season and other harsher environmental conditions, including a decrease in soil moisture and an increase in cold (*McCune & Johns, 2007*; *Rozema et al., 1997*). Flavonoids and terpenoids both play a part in a number of stress-response pathways (*Winkel-Shirley, 2002*; *Chappell, 1995*). The pattern observed in the gene expression of relevant metabolic enzymes suggests that these classes of chemicals are also likely to contribute to biological and chemical activity in *Sorbus* spp.

In addition, coastal areas undergo slower changes in temperature due to proximity to bodies of water, but are more exposed to physical factors such as wind. *Stushnoff & Junttila (1986)* identified a number of tree species (including *Sorbus aucuparia*) that hardened more quickly with respect to drops in temperature inland as compared to coastal areas. More recent work showed that trees can respond differently to the freezing process and accordingly to their local environments, via various physiological cues followed by phytochemical responses (*Li et al., 2003*). This study also showed significant differences in phytochemical concentration between southern and northern ecotypes (*Li et al., 2003*). Overall, our results show the predicted trend of greater gene expression, higher antioxidant capacity and greater phenolic content in coastal samples compared to inland. Yet, likely due to our limited number of samples and coarse sampling design (i.e., we acknowledge that "coastal" and "inland" designation of sites does hide a lot of fine scale spatial heterogeneity), these results will have to be looked at in much finer details in future studies.

Compared to squalene synthase, flavonol synthase was consistently expressed at a greater rate among species, locations and tissues. The reasons for this trend are unknown. The role of squalene synthase can also be seen in the production of sterols, which are known to help cell membrane fluidity and permeability (*Hartmann, 1998*). Whether it would explain the disparity between flavonol and squalene synthase expression is yet unknown. Nonetheless, it may also suggest that the flavonoids play a greater part in plant defences against environmental stressors in northern areas. This could be further pursued through quantitative phytochemical analyses. The lower expression of flavonol synthase observed in bark tissue compared to leaf still exceeds that of squalene synthase. As the bark of *Sorbus* is traditionally used for medicinal purposes, both classes could be significant contributors to the antioxidant and antidiabetic action observed (*Leduc et al., 2006*; *Spoor et al., 2006*; *Fraser et al., 2007*; *Vianna et al., 2011*; *Nachar et al., 2013*).

The greater expression of flavonol synthase compared to squalene synthase observed that we observed in the leaves of both *S. decora* and *S. americana* may reflect the fundamental role of flavonoids in defending against both UV-B damage and insect herbivory (*Rausher, 2006*; *Rozema et al., 1997*). This enzyme, suggested as crucial in a plant's adaptation to environmental cues and stresses (*Winkel-Shirley, 2002*; *Chaves, Escudero & Gutierrez-Merino, 1997*), is a major part of the plant's response to UV-B radiation, drought and

extreme temperature, all of which affect leaf structure more than bark (*Chaves, Escudero & Gutierrez-Merino, 1997*). The greater concentration of squalene synthase in the bark tissues of *S. decora* agrees with other findings that triterpenes, including those commonly identified within *Sorbus,* are often found in large concentration in the form of a resin within the inner bark (*Guerrero-Analco et al., 2010*; *Hanson, 2003*; *Theis & Lerdau, 2003*). While neither of the genes were found to be expressed significantly higher in the respective tissues of interest, we believe it is important to report these results as they demonstrate the utility of quantitative rtPCR in studying multiple genes from distinct tissues of plant of ethnobotanical, specifically medicinal, interest, and of this protocol for future studies into phytochemical evolution and action.

While several environmental factors have a major impact on secondary metabolite expression and accumulation, they should not be considered static forces. Concern as to the effect of environmental stressors such as ozone and acid rain has elicited studies that show that while acid rain has no observable effect on some secondary metabolites, increasing amount of UV-B radiation results in differences in tannin production and likely changes in the phenolic biosynthetic process (*Bassman, 2004*; *Jordan et al., 1991*). Furthermore, increasing temperatures in northern regions are likely to have many impacts on the flora adapted to the region, as well as the fauna they interact with, with likely metabolic, and possibly medicinal, consequences (*Ayres, 1993*).

Our results, in conjunction with ethnobotanical data showing that the bark of mountain ash is primarily used for medical purposes, indicate that both flavonoids and terpenoids could offer benefits to treat a number of T2D symptoms (*Vianna et al., 2011*). *S. decora*, with a larger and more coastal range, is likely to be a more important source of these compounds, as it is adapted to a greater variety of environmental stressors. Gene expression analyses are only one of many novel tools available to understand the forces contributing to diversity in plants used by humans and to facilitate development of treatments for modern diseases (*Cordell, 2011*). Used along with chemical quantification and other methods, these approaches can better identify how and why plants create beneficial compounds, as well as demonstrate the efficacy and accuracy of traditional medicinal methods in treating diseases of the past, present and future.

## ACKNOWLEDGEMENTS

AB, JAGA, AS, PH, JTA, TJ and AC were part of the CIHR Team in Aboriginal Antidiabetic Medicines. We extend very special thanks to Cree elders, healers and community members from Eeyou Istchee who kindly participated in the CIHR Team in Aboriginal Antidiabetic Medicines project. Their trust has enabled a useful exchange between Indigenous knowledge and Western science. We also thank the elders who reviewed and approved this work for publication: Smally and Laurie Petawabano, Johnny and Charlotte Husky Swallow, Eliza Mamianskum, Suzanne Atchynia, Andrew Kawapit, Josephine Diamond, Jimmy Trapper, Alex Weistche, and Lawrence and Yvonne Neeposh. We also appreciate the logistical support supplied by the Cree Board of Health and Social Services of James Bay (CBHSSJB). Thanks go to A Downing, N Roy, A Léger, F Landry, J Singh, C Ide, P Dufresne, J Bede, S Bailleul and S Daigle for technical and other forms of assistance.

### Funding

This work is supported by a Canadian Institute for Health Research grant awarded to TAAM (Haddad, Cuerrier, Johns), the Natural Sciences and Engineering Research Council (NSERC) (Discovery Grant to T Johns) and funds awarded to A Bailie from McGill University. The funders had no role in study design, data collection and analysis, decision to publish, or preparation of the manuscript.

### Grant Disclosures

The following grant information was disclosed by the authors:
Canadian Institute for Health Research.
Natural Sciences and Engineering Research Council (NSERC).
McGill University.

### Competing Interests

The authors declare there are no competing interests.

### Author Contributions

- Anna Bailie performed the experiments, analyzed the data, wrote the paper, prepared figures and/or tables, reviewed drafts of the paper.
- Sebastien Renaut analyzed the data, prepared figures and/or tables, reviewed drafts of the paper.
- Eliane Ubalijoro, José A. Guerrero-Analco and Ammar Saleem performed the experiments, reviewed drafts of the paper.
- Pierre Haddad, John T. Arnason and Timothy Johns conceived and designed the experiments, contributed reagents/materials/analysis tools, reviewed drafts of the paper.
- Alain Cuerrier conceived and designed the experiments, contributed reagents/materials/analysis tools, wrote the paper, reviewed drafts of the paper.

### Field Study Permissions

The following information was supplied relating to field study approvals (i.e., approving body and any reference numbers):

Field permit was covered within a Research Agreement among universities, the Cree Board of Health and Social Services of James Bay, and the participating communities (see http://www.taam-emaad.umontreal.ca/about%20us/CIHR-TAAM_Final_Research_ Agreement_signed_091030.pdf).

### Data Availability

The raw data has been supplied as Supplementary File.

### Supplemental Information

Supplemental information for this article can be found online at http://dx.doi.org/10.7717/ peerj.2645#supplemental-information.

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
