# Peer review of "Phytogeographic and genetic variation in Sorbus, a traditional antidiabetic medicine—adaptation in action in both a plant and a discipline"

_PeerJ, doi:10.7717/peerj.2645_

## Round 0.1 · original submission · Major Revisions

This is a very interesting study, using indigenous knowledge of medicinal plants and the known geographical variation in their efficacy to guide measurements of antioxidants and genes encoding enzymes involved in secondary metabolite synthesis. The paper was submitted with prior reviews from Axios. Overall, I think the authors have answered the points raised by those reviewers. Nonetheless, I think further revision is required. In particular, I’m not sure why so much is made of the observed differences between absolute abundance of squalene synthase and flavanol synthase genes unless it is established that transcription accurately predicts protein abundance and enzyme activity. If this information is available for the species examined, then it should be cited. If not, then I think the authors need to measure enzyme activity or, as they point out themselves in line 351, measure phytochemical levels, to justify the detailed discussion and speculation. Without this additional information, the only really valid results are the antioxidant levels, and I’m not sure that is enough.

---

## Round 0.2 · accepted · Accept

The additional citations and text clarifications address my concerns adequately.

External reviews were received for this submission. These reviews were used by the Editor when they made their decision, and can be downloaded below.